# Central Sensitization and Chronic Pain Personality Profile: Is There New Evidence? A Case-Control Study

**DOI:** 10.3390/ijerph20042935

**Published:** 2023-02-08

**Authors:** Marina Lopez-Ruiz, Andrea Doreste Soler, Jesus Pujol, Josep-Maria Losilla, Fabiola Ojeda, Laura Blanco-Hinojo, Gerard Martínez-Vilavella, Teresa Gutiérrez-Rosado, Jordi Monfort, Joan Deus

**Affiliations:** 1HM Hospital Sant Jordi, 08030 Barcelona, Spain; 2MRI Research Unit, Department of Radiology, Hospital del Mar, 08003 Barcelona, Spain; 3Centro de Investigación Biomédica en Red de Salud Mental, CIBERSAM G21, 08003 Barcelona, Spain; 4Department of Methodology, Faculty of Psychology, Autonomous University of Barcelona (UAB), 08193 Barcelona, Spain; 5Rheumatology Service, Hospital del Mar, 08003 Barcelona, Spain; 6Department of Clinical and Health Psychology, Universitat Autònoma de Barcelona, Bellaterra, 08193 Barcelona, Spain

**Keywords:** osteoarthritis, fibromyalgia, central sensitization, personality

## Abstract

Background: Personality traits are relevant for pain perception in persistent pain disorders, although they have not been studied in depth in sensitized and nonsensitized patients with knee osteoarthritis (OA). Objective: To explain and compare the personality profile of patients with OA, with and without central sensitization (CS), and fibromyalgia (FM). Setting: Participants were selected at the Rheumatology Department in two major hospitals in Spain. Participants: Case-control study where the sample consists of 15 patients with OA and CS (OA-CS), 31 OA without CS (OA-noCS), 47 FM, and 22 controls. We used a rigorous and systematic process that ensured the sample strictly fulfilled all the inclusion/exclusion criteria, so the sample is very well delimited. Primary outcome measures: Personality was assessed by the Temperament and Character Inventory of Cloninger. Results: The percentile in harm-avoidance dimension for the FM group is higher compared to OA groups and controls. The most frequent temperamental profiles in patients are cautious, methodical, and explosive. Patients with FM are more likely to report larger scores in harm-avoidance, with an increase in logistic regression adjusted odds ratio (OR_adj_) between 4.2% and 70.2%. Conclusions: Harm-avoidance seems to be the most important dimension in personality patients with chronic pain, as previously found. We found no differences between OA groups and between sensitized groups, but there are differences between FM and OA-noCS, so harm-avoidance might be the key to describe personality in patients with CS rather than the presence of prolonged pain, as found in the literature before.

## 1. Introduction

It is known that personality, a person’s predispositions and patterns of thinking, feeling, and acting [1], plays a very important role in humans. Its main characteristics are that it predicts important outcomes and is stable across the lifespan. Thus, personality traits manifest in a wide selection of behaviors both across contexts and over time [2]. It is easy to understand that personality will affect health, including pain. We already know that personality affects cancer or heart disease risk [1], smoking [3], patients’ satisfaction and clinical outcome after total knee arthroplasty [4], or quality of life [5], among others.

We would like to focus on two pathologies where chronic pain (CP) is the main feature. The first is osteoarthritis (OA), which is already one of the ten most disabling diseases in developed countries. The worldwide estimates are that 9.6% of men and 18% of women aged over 60 years have symptomatic osteoarthritis [6]. The second is fibromyalgia (FM), which is one of the most common CP conditions and probably the most famous. FM affects an estimated 3–6% of the world’s population, mostly women, rising in incidence with age [7]. Even if it is obvious that OA and FM are quite different, beyond chronic pain, there is a feature where both pathologies connect the sensitization. In particular, we focus on central sensitization (CS), which is defined as pain hypersensitivity, tactile allodynia, pressure hyperalgesia, aftersensations, and enhanced temporal summation [8]. In FM, the sensitization is one of the main characteristics [9,10] characterized by the dysfunction of neurocircuits, among them the perception, transmission, and processing of afferent nociceptive stimuli [11]; whereas in OA, sensitization tends to occur later in the disease and it is present in some patients but not in others [12,13] and may explain the difference in severity and resistance to analgesic treatment [14].

Psychological and personality factors should be considered an important contributor in the processing of CP, as we know that they play a role in modulating therapeutic responses and pain perception [15,16]. Personality has been a focus of research to understand its influences both on other psychological factors and on medical diseases. One theory of personality, which has neurobiological correlations, is the Biological Personality Theory of Robert Cloninger. This theory postulates that there are four dimensions of temperament (novelty-seeking: NS, harm-avoidance: HA, reward dependence: RD, and persistence: P) and three dimensions of character (self-directedness: SD, cooperativeness: C, and self-transcendence: ST). Temperament is stable and heritable; character can flow and be learned. In fact, FM is one of those pathologies where personality has been studied. Following the theory of Cloninger, it seems that there is a personality profile for FM: high scores of HA and low scores of SD [17,18,19]. Thus, FM patients are predisposed to acquire conditioned responses of avoidance to adverse stimuli, or potential danger, and are also inflexible or unable to adapt to the situation, compared to control subjects. Low scores of NS [17,20] have also been found in many CP diseases, not only in the personality profile of FM. To clarify the amount of outcomes in this issue and look for a common denominator in FM personality traits, Conrad et al. [21] made a review and concluded that high HA combined with low SD contributes to chronicity of pain, and that it is associated with avoidant coping strategies and catastrophic interpretation of signals of danger and punishment.

Little is known in the field of personality in OA, although it is the most prevalent musculoskeletal disease; presumably, plenty of OA patients suffer CP, and there are studies that show that the patient’s personality predicts recovery after total knee arthroplasty [22]. Çidem, Rezvani, and Karacan [23] studied personality in patients with knee OA, defined as affective temperament (depressive, hyperthymic, cyclothymic, irritable, and anxious), which describes the attitudes and behavior of individuals based on the constitutional, genetic, and biological factors. In knee OA patients, depressive temperament (hypercritical or complaining, passive or indecisive, self-critical, pessimistic, incapable of fun, and becoming worried) was the most common dominant affective temperament. However, they found no correlation between pain alleviation by physical therapy and dominant affective temperament, so no relationship between pain and personality. Many studies agree that there is a CP personality [21,24,25,26]; however, to the best of our knowledge there are no studies where CS personality traits are investigated, even when there is a growing tendency to somehow accept the existence of a personality of chronic pain patients [27]. Therefore, the aim of this study is to determine the temperament and character profiles of patients with knee OA, with and without CS, and compare them to the temperament and character profiles of FM patients as a prototypical central sensitization syndrome, and also to study the temperamental profile of patients with OA and FM to know how often it could become a personality disorder. Cloninger [28,29] explains personality disorders defined by temperamental profiles when the SD and CO character dimensions score low or below the 33–35th percentile, suggesting a lack of self-control and problems adjusting with the defined temperamental profile. Several meta-analyses have studied the effect size of the correlation coefficients between each of the temperament/character traits and personality disorder symptoms [30]. Based on the above information, it has been hypothesized that the personality profile in different chronic pain pathologies is mediated by the presence of central sensitization syndrome.

## 2. Materials and Methods

### 2.1. Participants

The initial sample was composed of 90 patients with knee OA diagnosis and 150 with FM diagnosis at the Rheumatology Department in two major hospitals in Spain (Hospital CIMA Sanitas and Hospital del Mar), selected by a senior rheumatologist and a senior clinical psychologist during an 18 month period. There was a control group with 35 participants.

Patients with OA were divided into 2 groups: (1) presence of clinical central sensitization, defined by the presence of both spreading sensitization and temporal summation to repeated pressure pain stimulation [8,12,31,32,33] (OA-CS group); and (2) absence of CS (OA-noCS group).

The principal inclusion criteria for the OA patients were:Radiological and clinical diagnosis of knee OA based on the American College of Rheumatology (ACR) criteria, affecting at least one knee with a minimum of 3 months of symptom duration prior to screening.Male or female (nonchildbearing potential) at least 45 years old.A minimum of 4 out of 10 on the numerical rating scale (item 5 of Brief Pain Inventory) at screening and/or a requirement for the use of an analgesic for the knee pain.

The specific inclusion criteria for the OA-CS group were:Clinical evidence of pain or altered sensations spread beyond the knee joint by manual palpation in the baseline rheumatologist assessment.At least 3 tender points in the extended version of the Arendt–Nielsen peripatellar map (excluding points 3, 7, and 8, which are part of the joint itself). A tender point is defined as a point showing a pressure pain threshold below 4 kg/cm^2^ [34].A pain score of 4 points or more in an 11-point verbal scale during a 2 s, 4 kg/cm^2^ pressure stimulation on the anterior surface of the tibial bone.Presence of temporal summation (increase of more than 1 point in an 11-point verbal scale after 10 repeated pressure stimulation at 1 s interstimulus intervals) on the most sensitive site of the peripatellar region [11].

The principal inclusion criteria for FM patients were:Diagnosis of FM following the ACR criteria [34].History of widespread nonarticular pain with insidious onset primary over 3 months.One year minimum of disease evolution.Absence of comorbid chronic fatigue syndrome.

Inclusion criteria for control group were:No history of rheumatic disorder, no history of functional pain or physical widespread pain.No history of Axis I or II psychiatric illness and no history of neurological disease.

Additionally, control participants underwent a brief semistructured clinical psychopathological interview according to DSM-5 diagnostic criteria carried out by a senior member of the research team (JD) to rule out any personality disorder. A convenience sampling was carried out where the researcher selects the patients who are easily accessible in a waiting room, through a satisfaction survey of the health system.

In all groups, patients with a history of psychotic disorder or substance abuse, patients with a history or diagnosis of personality disorders, and patients with a history of neuropathic pain were not included. A flow chart of the sample selected is shown in Figure 1.

The participants signed informed consent to accept the conditions of the study. We used a methodical and strict process that allowed us to ensure the sample strictly fulfilled all the inclusion/exclusion standards; therefore, the sample was very well delimited. The final sample (Table 1) was made up of 15 patients with OA and CS (OA-CS) aged between 44 and 81 (mean 66.37 years ± 8.77), 31 with OA without CS (OA-noCS) aged between 46 and 79 (mean 66.8 years ± 7.39), and 47 FM patients, without fatigue chronic syndrome aged between 32 and 63 (mean 46.47 years ± 7.9) and the percentage of low back pain (86%), migraine (76.7%), and irritable bowel syndrome (48.8%) as other manifestations of central sensitization syndrome. The control group was formed of 22 participants with a mean of 51.56 years (±11.41). This study was approved by the Local Ethics Committee and was in compliance with the Helsinki Declaration.

### 2.2. Procedure

First, patients attended rheumatologic visits where they were selected, and after verifying the inclusion/exclusion criteria as well as the will to participate, they were enrolled. During the next few days, patients attended psychological assessment, performed by the same clinical psychologist, which lasted about 90 min. Patients were visited twice in case of causing too much fatigue that may have influenced answers. The research only analyzes some aspects of this broad protocol.

### 2.3. Assessment

Revised Temperament and Character Inventory (TCI-R) [28]: We administered the Spanish version of the TCI-R [29]. The original TCI-R is a 240-item personality self-report questionnaire, based on Cloninger’s multidimensional and psychobiological model that accounts for both normal and abnormal variations in personality. TCI-R assesses 7 personality scales, each one with its own facets and a 5-point Likert response format (from 1 = definitely false to 5 = definitely true). All items are randomly listed and about half of them are reverse scored. The 4 temperament scales are:Novelty-seeking (NS), defined as the inclination to respond impulsively to novel stimuli with active avoidance of frustration.Harm-avoidance (HA) as the tendency to inhibit responses to aversive stimuli leading to avoidance of punishment.Reward dependence (RD) as the predisposition to answer to signals of reward.Persistence (PS) as the tendency to perseverance despite frustration and fatigue.

The 3 character scales are:Self-directedness (SD), which refers to the ability to control, regulate, and adapt behavior to fit the situation.Cooperativeness (CO), which is related to acceptance of other people.Self-transcendence (ST), which is viewed as the identification with everything conceived as essential and consequential parts of a unified whole.

The temperament facets in NS are:Exploratory excitability (NS1), impulsiveness (NS2), extravagance (NS3), and disorderliness (NS4).HA: Anticipatory worry (HA1), fear of uncertainty (HA2), shyness with strangers (HA3), and fatigability (HA4).RD: Sentimentality (RD1), openness to warm (RD2), attachment (RD3), and dependence (RD4).PS: Eagerness of effort (PS1), work-hardened (PS2), ambitious (PS3), and perfectionist (PS4).

The character facets for SD are:Responsibility (SD1), purposefulness (SD2), resourcefulness (SD3), self-acceptance (SD4), and enlightened second nature (SD5).CO: Social acceptance (C1), empathy (C2), helpfulness (C3), compassion (C4), pure-hearted conscience (C5).ST: Self-forgetful (ST1), transpersonal identification (ST2), and spiritual acceptance (ST3).

Specifically, this inventory allows us to assess the possibility of the psychometric presence of some symptomatic personality disorders (histrionic, narcissistic, avoidant, antisocial, borderline, schizoid, and obsessional) if the scores of SD and CO are low or very low (17–33%). The lower cut-off point is situated in 35 points and the upper in 75. This questionnaire has proved to be reliable in its original version, with internal consistency alpha coefficients ranging from 0.65 for persistence to 0.89 for cooperativeness. The Spanish version also shows good estimates of internal consistency alpha coefficients, such as in our study except in the novelty-seeking scale (Table 2).

### 2.4. Data Analysis

We performed nonparametric Kruskal–Wallis and chi-square statistical tests to analyze the differences between the four groups (OA-CS, OA-noCS, FM, and control group) in TCI-R mean percentile scores and in the distributions of the temperamental profiles and personality disorders of TCI-R, respectively. Logistic regression models were adjusted to identify the most characteristic personality pattern of these groups. For the statistical analysis we used the statistics Package SPSS 20 (SPSS Inc., Chicago, IL, USA).

## 3. Results

Descriptive features of the final sample are shown in Table 1.

Figure 2 and Table 2 show the mean percentile scores of the four groups of participants (OA-CS, OA-noCS, FM, and C) regarding both temperament and character dimensions of TCI-R. Most of the mean percentile scores ranged between the 40th and 60th percentile, but HA in FM patients was statistically higher than in the other groups (*p* < 0.001). Figure 3 and Table 3 show the distribution of temperamental profiles and personality disorders of the four groups in TCI-R. There are statistically significant differences between groups in the distribution of temperamental profiles (chi-square = 61.116, *p* < 0.001) but not in the distribution of personality disorders (chi-square = 24.261, *p* = 0.147). There are also no differences between groups in the frequency of absence/presence of personality disorder (Table 3).

After the contingency table and the chi-square test, we found a relation statistically significant between group (OA-CS, OA-noCS, FM, and C) and temperamental profile. Figure 3 and Table 3 show the frequency of temperamental profile (cautious and methodical) (chi-square *=* 61.116, *p* < 0.0005) and personality disorder (chi-square *=* 24.261, *p* < 0.147) obtained depending on the temperamental profile. The contingency coefficient scored 0.589 (*p* < 0.05), which is high (according to the formula: Max (C) = √χ^2^/(χ^2^ + n)) and is directly proportional. Figure 4 suggests, psychometrically, the presence/absence of personality disorders.

To obtain a descriptive profile of personality for each group, a logistic regression analysis was completed. The initial model included the following confounding factors: gender, age, academic level, and cognitive screening.

### OA Patients versus Controls and FM Patients versus OA-noCS Patients

Table 4 summarizes the results of the logistic regression model that investigates the differential TCI-R personality profile of OA patients versus control subjects. In the reward dependence dimension, OA patients are more likely to report larger scores, showing statistical significance (increase in ORadj between 7.9% and 15.3%).

Table 4 also summarizes the results of the logistic regression model that investigates the differential TCI-R personality profile of FM patients versus OA-noCS patients. Patients with FM are more likely to report larger scores in the harm-avoidance dimension of temperament (increase in ORadj between 4.2% and 70.2%), showing statistical significance.

The outcomes of the logistic regression between OA-CS and OA-noCS and between FM and OA-CS show no statistical significance in their profiles. We also performed logistic regression in every subdimension of each temperament, and character dimensions where we found no significant differences.

## 4. Discussion

The aim of this research was to describe the most frequent temperamental profile for different groups of patients, including the control group, to study whether the participants show a potential personality disorder or not, and of what type, and finally to describe the pattern of personality for different groups, focusing on the presence of CS. Many researchers found higher scores in HA and lower in SD, with some variations with the role of the other dimensions of temperament and character, for example, lower scores in NS [15,20]. Gencay-Can and Can [18] found significant higher scores in ST in FM patients compared to control subjects. Leombruni et al. [19] found two types of FM patients according to personality traits and clinical features. Those in cluster 1 show higher vulnerability to health, anxiety, mood, and personality problems. Those in cluster 2 show better quality of life, less psychiatric and organic symptoms, and lower severity. In our study, FM patients showed the highest score in the HA dimension which is a different characteristic from the other groups. Additionally, in these patients, the dimension of SD shows the lowest score; however, this is without statistical significance.

Regarding the temperamental profile results of combining temperamental dimensions NS, HA, and RD, for OA-noCS patients, the most frequent profiles are cautious (19.4%) and explosive (19.4%), for OA-CS it is methodical (26.7%), for FM patients it is cautious (36.2%), and for controls it is independent (36.4%). Explosive (high scores in NS), methodical, and cautious (both low scores in NS) show high scores in the HA dimension, while independent shows high scores in RD. When SD and CO are low, the possibility of having a personality disorder exists. In our sample, most of the patients do not show a potential disorder. However, patients who do show obsessive, avoidant, and borderline disorders correspond to methodical, cautious, and explosive temperamental profile. Thus, patients with a potential personality disorder of cluster C show high scores in HA. Thus, both patients with and without personality disorder show inhibition, passive avoidance, pessimism, fear of uncertainty, cautiousness, shyness, anticipation of problems, and fatigue after stress. When the disorder is present, these traits are more exaggerated, meaning that the person is not able to control his or herself driven by the situation.

After the regression analysis, we found that patients with and without CS are more likely to show RD traits, a fact that has not been found in research before. These dimensions assess the tendency to be influenced by others, be very sensitive to social signs, sentimental, and be able to understand the feelings of others. Garcia-Fontanals et al. [17] and Gencay-Can and Can [18] found that FM patients are different from control subjects in NS, HA, and SD and in HA, SD, and ST, respectively. As OA patients from our sample are different in only one dimension, we could assume that they are less likely to suffer personality disorders than FM patients, who show differences in three dimensions, including SD, one of the most relevant dimensions regarding personality disorders. On the other hand, the pattern of FM patients compared with OA-noCS shows higher scores in HA. Other differentiated patterns, meaning OA-CS versus OA-noCS and FM versus OA-CS, showed no diverse patterns. Thus, if there are no differences, either between groups with OA, or groups with CS, but there are differences between FM and OA-noCS, we could assume that the HA dimension is the key. In Çidem et al.’s [23] knee OA research where they studied the personality profile, they found that the depressive temperament was the most frequent for these patients. This temperament has common features with the HA dimension, for example, being worried, passive, pessimistic, and anxious. In addition, in Torres et al. [35], they compared personality profiles assessed by the NEO Five-Factor Inventory (NEO-FFI-R) of patients who suffer FM, FM and comorbid rheumatologic disorder, CP disorder, and patients with drug-resistant epilepsy as a sample of nonpainful chronic illness. They did not find any differences in the personality profiles of FM patients compared to other groups of patients. In our opinion, perhaps there are no differences between patients with CP, as the classical literature states over the years [21,24,26], but there are differences between patients with or without CS.

There are authors who state that central sensitization syndrome has been produced by the existing duration of chronic pain that begins in the early stages of life or due to innate personality traits (temperament), learned personality traits (character), or a combination of both. Our study covers an older population and, therefore, innate temperament traits are what determine the presence of SSC [36].

This research has some limitations:The sample size is small, especially for the OA-CS group. Given the size of the sample, we were not able to carry out an analysis that differentiates between gender and age.External validity is limited by the sample size, even though the sample is very specific and restrictive, with the inclusion/exclusion criteria. For example, FM patients showed no chronic fatigue, which is a very common characteristic.The FM group was not divided into potential subgroups of patients according to severity; for example, there are few studies that show evidence in this field. Thus, it is complicated to find previous studies to compare our results with others.The effects of other symptoms of central sensitization that may be present in these patients, such as migraine, irritable bowel syndrome, or low back pain, have not been studied in OA groups due to the small sample size.

Therefore, our results show that patients with CP show a diverse personality pattern compared to control subjects [27,37], and it is characterized by the HA dimension. However, the presence of CS seems to determine the importance of this dimension, because in CS groups it is a differential feature, but not in CP patients, which means the OA group.

Multiple differences between groups with CS and without CS have been reported in previous research; Clark et al. [38] found a high prevalence of extreme trait sensory hyper, hyposensitivity profile score and defensive high anxious, related with catastrophism, rumination, magnification, and helplessness found in CS patients [39].

Personality assessment with TCI-R gives us the opportunity to evaluate treatment responses. Many studies found that the pretreatment level in the SD dimension predicts the scope of improvement. When the therapy is successful, levels of SD increase, giving control across the situations [15,21]. Therefore, personality assessment is useful to describe the personality pattern in the clinical practice and its changes, and this is why we encourage researchers, clinicians, and ourselves to give it an important role in the regular treatment of these kind of patients, especially those who suffer from CP and SC.

We changed the focus from CP to CS; however, more investigations are needed in CS syndromes to find out if this phenomenon is an important factor in the role of the HA dimension, and not the presence of CP, as is normally reported. Further research is recommended to include the controlled pain parameter that differs in personality traits.

## 5. Conclusions

In conclusion, harm-avoidance seems to be the most important dimension in personality patients with chronic pain, as previously found. We found no differences between OA groups and between sensitized groups, but there are differences between FM and OA-noCS, so harm-avoidance might be the key to describe personality in patients with CS rather than the presence of prolonged pain, as found in the literature before.

## Figures and Tables

**Figure 1 ijerph-20-02935-f001:**
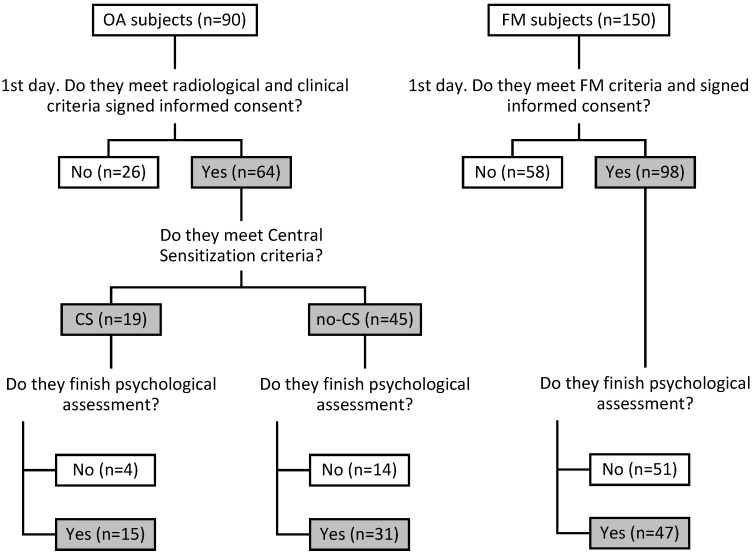
Flow chart of the selection sample process. Note: OA: osteoarthritis, FM: fibromyalgia, OA-CS: osteoarthritis with central sensitization, OA-noCS: osteoarthritis without central sensitization.

**Figure 2 ijerph-20-02935-f002:**
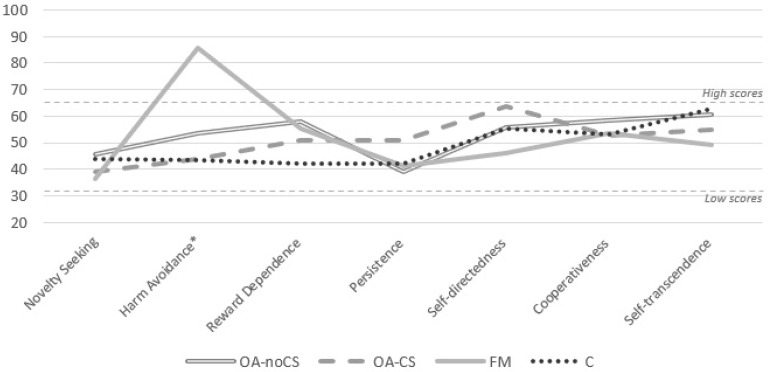
Group profiles of mean percentile scores of temperament and character dimensions of TCI-R. Note. TCI-R: Temperament and Character Inventory; OA-CS: osteoarthritis with central sensitization; OA-noCS: osteoarthritis without central sensitization; FM: fibromyalgia; C: control group; SD: standard deviation; * *p* < 0.001.

**Figure 3 ijerph-20-02935-f003:**
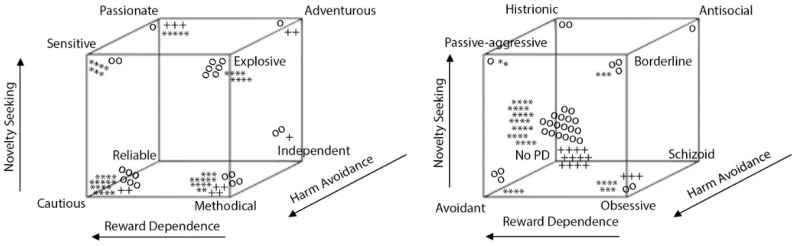
Temperamental profile and personality disorders in TCI-R for different groups of patients. Note. TCI-R: Temperament and Character Inventory, +: osteoarthritis with central sensitization, o: osteoarthritis without central sensitization, *: fibromyalgia. Multiple *, o, + implies greater number of cases. PD: personality disorders (histrionic: excessive attention-seeking behavior; passive-aggressive: procrastination, covert obstructionism, inefficiency and stubbornness; avoidant: avoidance to feared stimuli; obsessive: excessive need for orderliness, neatness, and perfectionism; schizoid: lack of interest in social relationships; borderline: extreme moods, fluctuating self-image, and erratic behaviors characterized by impulsive actions; antisocial: pervasive and persistent disregard for morals, social norms, and the rights and feelings of others.

**Figure 4 ijerph-20-02935-f004:**
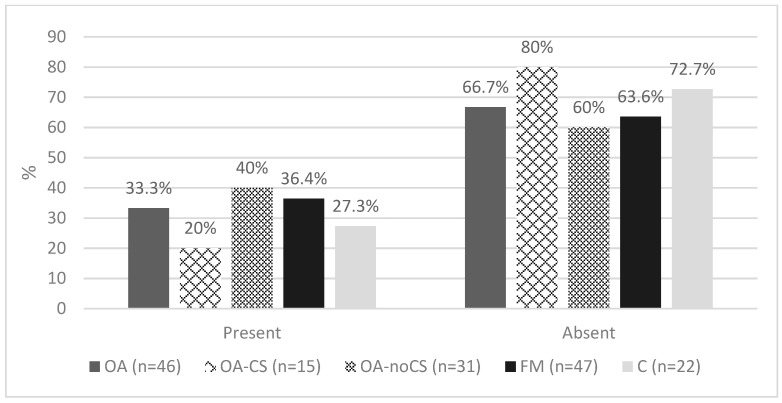
Presence/absence of TCI-R personality disorders by group. Note. TCI-R: Temperament and Character Inventory, OA: osteoarthritis, OA-CS: osteoarthritis with central sensitization, OA-noCS: osteoarthritis without central sensitization, FM: fibromyalgia, C: control.

**Table 1 ijerph-20-02935-t001:** Sociodemographic and clinical characteristics of the sample.

	Groups				
Factors	OA	OA-CS	OA-noCS	FM	Control
N	46	15	31	47	22
Gender (women %)	71.7	84.2	65.9	100	59.3
Age ($\bar{x}$(SD))	66.67 (7.78)	66.37 (8.77)	66.8 (7.39)	46.47 (7.92)	62.92 (7.39)
Months after diagnosis ($\bar{x}$ ± SD)	55.28 (63.34)	50.58 (54.09)	57.46 (67.71)	84.38 (54.14)	-
Educational level (%)					
Uneducated	6.7	10.5	4.9	0	0
Primary education	21.7	26.3	19.5	10.6	26.9
Secondary education	15	10.6	17.1	21.3	12.1
General education	5	10.5	2.4	12.8	7.4
Vocational education and training and/or Higher education	31.7	36.8	29.3	17	30.6
Bachelor and/or University Degree	20	5.3	26.8	38.3	23
Drug use (%)					
Painkiller	50	16	34	16	0
Non-steroidal Anti-inflammatory drugs (NSAIDs)	51	16	35	17	2
Antiepileptic drugs	2	0	2	4	3
Antidepressant drugs	14.9	15.8	19.5	85.7	7.4
Questionnaires					
FIQ	-	-	65.99 (14.01)	-	
Mini-Mental: 27.61 (2.68)	26.82 (3.17)	28 (2.36)	-	27.5 (2.45)	

OA: osteoarthritis, OA-CS: osteoarthritis with central sensitization, OA-noCS: osteoarthritis without central sensitization, FM: fibromyalgia, SD: standard deviation, FIQ: fibromyalgia impact questionnaire, Mini-Mental: OA patients carried out cognitive screening because of their advanced age and the possibility of cognitive impairment. FM patients did not need it due to their age and study objectives.

**Table 2 ijerph-20-02935-t002:** Percentile scores of temperament and character dimensions of TCI-R by group.

	Group					
TCI-R Dimensions	OA-noCS (*n* = 15) Mean (SD)	OA-CS (*n* = 31) Mean (SD)	FM (*n* = 47) Mean (SD)	C (*n* = 22) Mean (SD)	K–W Test *p*	*Cronbach’s Alpha*
Novelty-seeking	45.58 (27.53)	39.00 (29.25)	36.21 (22.33)	43.68 (24.25)	0.379	0.43
Harm-avoidance *	53.61 (30.30)	43.80 (30.44)	85.85 (15.17)	43.27 (30.29)	<0.001 *	0.77
Reward dependence	57.94 (22.80)	51.13 (24.62)	55.15 (27.89)	42.09 (32.94)	0.227	0.64
Persistence	39.03 (28.15)	50.8 (32.94)	41.13 (30.58)	42.05 (26.07)	0.675	0.77
Self-directedness	55.71 (28.03)	63.67 (31.40)	45.98 (30.27)	55.36 (29.63)	0.178	0.74
Cooperativeness	58.55 (30.84)	52.87 (32.39)	53.74 (27.88)	52.95 (29.05)	0.817	0.71
Self-transcendence	60.71 (30.03)	54.87 (35.32)	49.26 (30.07)	62.91 (32.82)	0.275	0.78

Note. TCI-R: Temperament and Character Inventory; OA-CS: osteoarthritis with central sensitization; OA-noCS: osteoarthritis without central sensitization; FM: fibromyalgia; C: control group; SD: standard deviation; K–W *p*: statistical significance of the nonparametric Kruskal–Wallis test; * statistically significant difference.

**Table 3 ijerph-20-02935-t003:** Distribution of TCI-R temperamental profiles and TCI-R personality disorders by group.

	Group				
TCI-R Temperamental Profile	OA (*n* = 46) *n* (%)	OA-CS (*n* = 15) *n* (%)	OA-noCS (*n* = 31) *n* (%)	FM (*n* = 47) *n* (%)	C (*n* = 22) *n* (%)
Methodical	8 (17.4)	4 (26.7)	4 (12.9)	14 (29.8)	3 (13.6)
Cautious	8 (17.4)	2 (13.3)	6 (19.4)	17 (36.2)	3 (13.6)
Explosive	6 (13)	0	6 (19.4)	8 (17.0)	2 (9.1)
Sensitive	2 (4.3)	0	2 (6.5)	7 (14.9)	1 (4.5)
Passionate	8 (17.4)	3 (20)	5 (16.1)	1 (2.1)	4 (18.2)
Independent	3 (6.5)	1 (6.7)	2 (6.5)	0	8 (36.4)
Adventurous	3 (6.5)	2 (13.3)	1 (3.2)	0	0
Reliable	8 (17.4)	3 (20)	5 (16.1)	0	1 (4.5)
Chi-square = 61.116, *p* < 0.001.					
**TCI-R Personality Disorders**	**OA (*n* = 46)** ** *n* ** **(%)**	**OA-CS (*n* = 15)** ** *n* ** **(%)**	**OA-noCS (*n* = 31)** ** *n* ** **(%)**	**FM (*n* = 47)** ** *n* ** **(%)**	**C (*n* = 22)** ** *n* ** **(%)**
Absent	30 (66.7)	12 (80)	18 (60)	28 (63.6)	16 (72.7)
Obsessive	5 (11.1)	3 (20)	2 (6.7)	7 (15.9)	1 (4.5)
Avoidant	3 (6.7)	0	3 (10)	4 (9.1)	0
Borderline	3 (6.7)	0	3 (10)	3 (6.8)	2 (9.1)
Passive–aggressive	1 (2.2)	0	1 (3.3)	2 (4.5)	0
Histrionic	2 (4.4)	0	2 (6.7)	0	0
Antisocial	1 (2.2)	0	1 (3.3)	0	3 (13.6)
Chi-square = 24.261, *p* = 0.147					

Note: TCI-R: Temperament and Character Inventory, OA: osteoarthritis, OA-CS: osteoarthritis with central sensitization, OA-noCS: osteoarthritis without central sensitization, FM: fibromyalgia, C: control, *p*: statistical significance.

**Table 4 ijerph-20-02935-t004:** Comparison between TCI-R personality profiles of patients with OA and the control group.

OA vs. C			
	OR_adj_ (95% CI)	(OR_adj_ − 1) × 100	*p*
Reward Dependence (RD)	1.079 (1.079–1.153)	7.9% (7.9–15.3%)	0.024
**FM vs. OA-noCS**			
	OR_adj_ (95% CI)	(OR_adj_ − 1) × 100	*p*
Harm Avoidance (HA)	1.332 (1.042–1.702)	33.2% (4.2–70.2%)	0.022

Note. OA: osteoarthritis, C: control, ORadj: logistic regression adjusted odds ratio, 95% CI: 95% confidence intervals, *p*: *p*-values, (ORadj − 1) × 100: shows percentage of change. The regression model included gender, age, and academic level. OA-noCS: osteoarthritis without central sensitization, FM: fibromyalgia.

## Data Availability

Neither of the experiments reported in this article were formally preregistered. Neither the data nor the materials have been made available on a permanent third-party archive. All data are stored in the corresponding file (paper and electronic) of the research group. For more information, please, contact the corresponding author.

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
