# Peer review of "Central Sensitization and Chronic Pain Personality Profile: Is There New Evidence? A Case-Control Study"

_ijerph, 2023, doi:10.3390/ijerph20042935_

Round 1

Reviewer 1 Report

Many clinical disorders freequently found in the population in relationship with their high prevalence are known confounding factors with regard to the personality trait.

None of these has been considered as a bias of this study, at least in the discussion.

Among these conditions we may include, for example, migraine and tension type headache, low back pain and irritable bowel sindrome, that are present in more than 15-20% of the population.

How many patients of this study cohort are migraineurs? How many have Irritable Bowel sindrome? At least in the discussion these confounders may be considered.

Author Response

The contributions are very interesting, and it is certainly an aspect that we have not delved into, so we have included it in the article in two sections. We have briefly discussed the symptoms in patients with FM but have not included it in the knee OA group because they have not been recorded as associated symptoms of central sensitization. Lines 167-169. Consequently, we have put as a limitation of the study that we have not been able to evaluate/analyse the effect of these symptoms/syndromes of central sensitization associated with these rheumatic pathologies. Lines 397-399.

Reviewer 2 Report

I am very happy to review this well presented and designed research article. I think this research have creative aims and hypotheses. Particularly, investigaton of the personality disorder features from the view of Cloninger’s personality theory including temperament and character dimensions is remarkable.  In the following, I declared my suggestions about this research.

Introduction section:

The introduction part is very well written and the background of the mentioned topics of the present study was presented sufficiently. I wish to mention a few points about this section.

In the introduction section, central sensitization issue may be explained with a few more sentences in order to understand the association between central sensitization and clinical features of the mentioned disorders ( fibromyalgia and osteoarthritis).

Also, to study the temperamental profile of patients with 91 OA and FM to know how often it could become a personality disorder. In line 91 and 92, for the sake of to clarify the purpose of this sentence and this sentence is related with the main aims of the study, in the last paragraph a few sentences may be added about the association between Cloninger’s temperament and character suggestions and personality disorders.

Materials and Methods section:

 The selection of the participants adn the course of the participants during the study is very well explained both with  words and figure.

The assessment materials and the assessment procedures are very well written.

Results section:

The results of the study is very well presented considering the aims of the study.

Discussion section:

Discussion section is very well and I do not have any offer considering this part of the present study.

Author Response

Thank you for the contribution. We have briefly explained what is the phenomenon of central sensitization in patients with fibromyalgia and in patients with osteoarthritis (Lines 54-59 and references 38 and 39).

We have included more information related to personality disorders according to the Cloninger model in Lines 99 to 193 and reference 39.

Reviewer 3 Report

The present study aims to explain and compare the personality profile of patients with osteoarthritis, with and without central sensitization and fibromyalgia. 

 The results highlighted that the percentile in the Harm-Avoidance dimension for the fibromyalgia group is higher compared to osteoarthritis groups and controls. Patients' most frequent temperamental profiles are Cautious, Methodical, and Explosive. 

I have several suggestions.

ABSTRACT:

1.     Please expands the ORadj abbreviations in the summary.

INTRODUCTION

2.     Please describe the mechanism of Central Sensitization and its relation to Fibromyalgia.

RESULTS:

3. Please indicate whether there were significant differences in terms of the gender of the participants. Could gender have influenced the results obtained in this study? Was the sample size calculated before the examination? Did the age of the subjects differ between the study groups? These aspects should be clarified in this paper.

OVERALL:

4. Please improve the style and language of the paper. I hope my suggestions will help improve this work. 

Author Response

Thank you for the correction, we have included the meaning of the abbreviation in the abstract.

Thank you for the contribution. We have briefly explained what is the phenomenon of central sensitization in patients with fibromyalgia and in patients with osteoarthritis (Lines 54-59 and references 38 and 39).

Thank you for the suggestion. We have included the sentence: Given the size of the sample, we have not been able to carry out an analysis that differentiates between gender and age in limitations. Lines 389-390.

We appreciate the suggestion that will help us improve the quality of the article. The article has been newly reviewed by a professional from this native English field.

Round 2

Reviewer 3 Report

Dear Authors,

Thank you very much for considering my comments. I would like the authors to correct the order of citations in the work in connection with adding new references.

Kind regards